Dynamics and stability of directional jumps in the desert locust

Gvirsman Omer 1 2
Kosa Gabor gkosa@post.tau.ac.il 1
Ayali Amir ayali@post.tau.ac.il 2
1 School of Mechanical Engineering, Faculty of Engineering, Tel Aviv University , Tel Aviv , Israel
2 Department of Zoology, Tel Aviv University , Tel Aviv , Israel
Diao Jiajie
Electronic publication date: 2016 Sep 28
Publication date: 2016
Volume: 4
Electronic Location ID: e2481
Received 2016 Jun 20; Accepted 2016 Aug 23
Copyright: ©2016 Gvirsman et al.
Copyright year: 2016
Copyright holder: Gvirsman et al.
License: This is an open access article distributed under the terms of the Creative Commons Attribution License, which permits unrestricted use, distribution, reproduction and adaptation in any medium and for any purpose provided that it is properly attributed. For attribution, the original author(s), title, publication source (PeerJ) and either DOI or URL of the article must be cited.
License URL: https://creativecommons.org/licenses/by/4.0/

Keywords: Trajectory control, Biomechanics, Flight initiation, Schistocerca gregaria, Angular velocity, Simulation

Funding: Pearls of Wisdom This study was supported by an award from Pearls of Wisdom, Israel. The funders had no role in study design, data collection and analysis, decision to publish, or preparation of the manuscript.

==============================
Locusts are known for their ability to jump large distances to avoid predation. The jump also serves to launch the adult locust into the air in order to initiate flight. Various aspects of this important behavior have been studied extensively, from muscle physiology and biomechanics, to the energy storage systems involved in powering the jump, and more. Less well understood are the mechanisms participating in control of the jump trajectory. Here we utilise video monitoring and careful analysis of experimental directional jumps by adult desert locusts, together with dynamic computer simulation, in order to understand how the locusts control the direction and elevation of the jump, the residual angular velocities resulting from the jump and the timing of flapping-flight initiation. Our study confirms and expands early findings regarding the instrumental role of the initial body position and orientation. Both real-jump video analysis and simulations based on our expanded dynamical model demonstrate that the initial body coordinates of position (relative to the hind-legs ground-contact points) are dominant in predicting the jumps’ azimuth and elevation angles. We also report a strong linear correlation between the jumps’ pitch-angular-velocity and flight initiation timing, such that head downwards rotations lead to earlier wing opening. In addition to offering important insights into the bio-mechanical principles of locust jumping and flight initiation, the findings from this study will be used in designing future prototypes of a bio-inspired miniature jumping robot that will be employed in animal behaviour studies and environmental monitoring applications.

Introduction

Locusts are extremely capable jumpers. Whether escaping predators or merely getting from one location to another, they are able to aim their jumps at specific points in space (Collett & Paterson, 1991; Santer et al., 2005; Sobel, 1990), reaching distances of up to 20 times their own body length (Bennet-Clark, 1975). In adult locusts the jump also serves in flight take-off (Katz & Gosline, 1993), propelling the insect into the air to allow the initiation of flapping flight. Understanding the details of locust jumping behaviour, and the way by which specific aspects of the behaviour affect the properties of the resulting jump, is critical for revealing the underlying bio-mechanical principles of locust jumping and flight initiation. Such knowledge could also serve in designing much sought-after small jumping robots (e.g., Zaitsev et al., 2015a; Zaitsev et al., 2015b), as the similar size-scale is also resulted in similar challenges and difficulties in achieving high-performance and high-accuracy jumps.

The key to successful jumps lies in the production of sufficient power (Gabriel, 1984). The locust enhances its power by storing energy in the hind-legs’ cuticle and soft tissues prior to the jump and releasing it simultaneously with muscle action (Bennet-Clark, 1975). The motor program and mechanics responsible for producing and controlling the thrust of the jump have been widely studied (Burrows, 1995; Heitler & Burrows, 1977). Only a relatively few studies, however, have addressed the issue of trajectory control. For any jump, the initial trajectory is defined by the magnitude, azimuth and elevation of the take-off velocity vector. The locust controls its azimuth by rotating its body towards the desired direction through rapid movements of the fore- and meso-thoracic legs (Santer et al., 2005; Sutton & Burrows, 2008), whereas elevation is separately controlled by establishing the position of the hind legs through their rotation at the thoraco-coxal (TC) and coxo-trochanteral joints, accelerating the body along a line connecting the distal end of the tibia and the proximal end of the femur (Sutton & Burrows, 2008). These actions (hereafter referred to as the aiming manoeuvres) are very rapid and manifested shortly before and during jump initiation, thus enabling the locust a hasty escape in an appropriate direction when surprised by a predator. Although asynchrony between the hind legs might appear to offer an intuitive strategy for controlling the jump trajectory, it is not exploited by the locust (Santer et al., 2005; Sutton & Burrows, 2008). While the force applied by the hind legs accelerates the locust’s body, if the force vector of each leg does not pass through the centre of mass (COM) of the body, it will additionally produce a torque that causes rotation. Rotational velocity during the air-born phase can lead to difficulties in flight initiation or in safe landing. Cofer et al. (2010) suggested that the locust uses two mechanisms to minimize pitch rotations, also known as tumbling. In the first, setting the pitch according to the elevation angle, the COM is brought in-line with the force vector, thus minimizing the thrust force torque; in the second, a counter torque is produced by way of contraction of the dorso-longitudinal muscles during the jump. Those authors also observed that tumbling locusts were biased to rotate their body in a head-upwards direction, and hypothesized that the reason for this bias could be to enhance lift during flight initiation.

While mechanisms of elevation control have been well explained, the mechanics underlying azimuth control are much less understood. Cofer et al. (2010) focused on straight jumps, where due to the symmetry between the hind legs’ position, the locust rotates almost only about the pitch axis. In directional escape jumps, however, the symmetry of the hind legs position is lost, leading to the production of torque about all three principal axes (yaw, pitch and roll), and therefore to the development of rotational velocity about these axes. The nature of these additional rotations (about the yaw and roll axes) and the locust’s means to control them have not been studied to date.

In the current study we further argue that azimuth, elevation and stability control are coupled problems in the sense that they cannot be explained independently. To the best of our knowledge previous reports (e.g., cited above) have not addressed all three issues simultaneously. We aim to provide a dynamic model capturing the full spatial mechanics of the jump, and to determine the locust’s strategies for controlling the jump trajectory and rotational instability.

To accomplish this aim, we extended an existing two degrees of freedom (DOF), point mass dynamic model (Sutton & Burrows, 2008) into a six DOF, rigid body model. To complement the extended model we observed and monitored adult locust jumps through synchronous multi high-speed video cameras, enabling extraction of the full six DOF trajectory of the locust body during the jump. By comparing the real jump trajectories to trajectories predicted by computer simulations we validated the dynamic model, enabling us to explore further the locust jump through simulated experiments.

Figure 1 From real jumps to simulation.

(A) A locust during an experiment prior to a jump. The white dots act as markers for the video tracking. (B) The locust coordinate system used to measure locust orientation and position with respect to the ground coordinate system (see text). (C) A diagram of the dynamic model of the locust mechanics—the body is represented as a cuboid upon which two forces (green arrows) act. The forces’ directions are set to be the same as a line connecting the contact point of the tibia with the ground and the connection point between femur and body. Equivalent points in the model and in the locust body are marked with yellow dots in (B) and (C).

Materials and Methods

Video monitoring of locust jumps

Individual adult female desert locusts (Schistocerca gregaria) were obtained from our breeding colony at Tel Aviv University. Each locust was weighed and three dots (markers) in white acrylic paint were drawn in a triangular formation on the dorsal side of the pronotum, to provide position markers for motion analysis (Fig. 1A). Locusts were positioned on a 9 × 5 cm platform covered with sandpaper, to minimize the chances of slipping, and were stimulated to jump by way of introducing fast moving objects into their visual field. For each locust a maximum of 10 jumps were recorded at minimal intervals of 10 min between jumps. The jumps were recorded at 2000 frames s−1 at a resolution of 1,024 pixels × 1,024 pixels by three synchronous Photron SA3 Fastcam video cameras (Photron, Inc., San Diego, CA, USA) with an exposure of 1/6000 s. Although only two cameras were needed to fully reconstruct the 3D position of the markers, a third camera was used to ensure that all markers were in sight of at least two cameras at all times, irrespective of the orientation of the locust body. A cube of known dimensions placed on the jumping platform was used for camera calibration via direct linear transformation, utilizing the open source DLTdv5 package (Hedrick, 2008) for MATLAB (Mathworks, Natick, MA, USA). Overall fifty jumps by 12 locusts were recorded.

Processing video data

Digitization of the markers and reconstruction of their position were performed using the DLTdv5 package (Hedrick, 2008). For each recorded jump the following procedure was performed: the three markers on the locust’s pronotum were tracked in all the frames using the automated tracking feature. The contact points of the distal end of the hind legs tarsi with the ground were digitized in one frame in which they were most easily identified. The TC joint was digitized in four frames evenly spaced through the jump duration. Following marker digitization and computation of the locust body position and orientation, the data were smoothed using a fourth-order, zero-lag, low-pass Butterworth filter with a cutoff frequency of 200 Hz. We mostly focused on the initial stages of the jump, when the extension of the hind legs was first observed, and on the time point of take-off, when the hind legs lost contact with the ground. By smoothing the raw data from 10 ms prior to jump initiation to 10 ms after take-off, endpoint errors associated with filtering were avoided. For further details regarding data processing see (Fig. S1).

Jump kinematics analysis

Two coordinate systems were defined to facilitate the kinematics computation (Fig. 1B). The first was a global coordinate system, namely the ground system, whose origin was located midway between the contact points of the hind legs with the ground (Og). It was oriented so that one of its axes was in the direction of a line connecting the contact points of the hind legs with the ground (yg), another axis was perpendicular to the jumping platform (zg), and a third axis was perpendicular to the first two axes, according to the right-hand convention (xg). The second coordinate system was a body-attached coordinate system, namely the locust system, with its origin located midway between the TC joints, and its axes coincident with the main axes of the locust body. The instantaneous location of the locust was determined by the position of the origin of the locust coordinate system (Ol), expressed in spherical coordinates (α-Horizontal angular movement; β-Vertical angular movement; r-Radial distance from the origin) with respect to the ground system, as illustrated in Fig. 1B. The instantaneous linear velocity was calculated by numerically differentiating the locust’s location. The instantaneous orientation of the locust system was described using the roll-pitch-yaw (Denoted by the angles ψ, θ, and ϕ, respectively) rotation convention, and the angular velocities were described about the main axes of the locust system (Denoted by the angles ψ ˙, θ ˙, and ϕ ˙, respectively. See Fig. 1B). Jump azimuth was the angle between xg and the projection of the linear velocity on the horizontal plane, and jump elevation was the angle between the linear velocity vector and the horizontal plane. Calculations relating the marker positions obtained from video analysis to the kinematic analysis are detailed in the supplementary data (Fig. S1).

Dynamical model

Sutton & Burrows (2008) approximated each locust hind leg as two connected homogenous rods, representing the femur and the tibia. Under the assumption that the hind legs are massless, they showed that each leg can only produce a force in the direction of a line connecting the distal end of the tibia and the proximal end of the femur. We have demonstrated that this holds true not only in 2D but also in 3D (see Appendix). To simplify further the model, the two-segment hind legs were replaced with equivalent forces (Fig. 1C). The locust body was approximated as a homogenous, rectangular cuboid upon which the forces representing the hind legs act (Fig. 1C). The effect of gravity and air resistance forces until take-off were assumed to be minor compared to the hind-legs’ thrust force, and are thus ignored. It is important to note that according to this model, the directions of the hind legs’ thrust forces are solely defined by the body’s relative position with respect to the hind legs’ ground contact points. Therefore, all the remaining locust’s DOF, such as head rotation, rotations at the different segments of the legs and abdomen flexibility have no effect on the jump’s trajectory. This highly simplified both, the video monitoring and computer simulations.

Simulations

The motion equations governing the dynamic model were derived using Maple (Maplesoft, Waterloo, ON, Canada), and solved with the MATLAB ODE45 solver. The main simplification for running the simulation was that aiming the jump is achieved solely by changing body posture prior to the jump, while there is no control factor during the jump itself. Each simulated jump was initialized using data obtained from the jump videos: contact points of the dorsal end of both hind-leg tarsi to the platform, distance between TC joints, initial body position and orientation angles, and jump duration. The mass was set according to the weight measurement. Two parameters could not be obtained from the video sequences: the position of the COM and the reaction forces exerted by the hind legs. The COM position with respect to Ol was set according to previous measurements (Taylor & Thomas, 2003). In locust jumps, the ground reaction force has a typical profile, starting at 0 at the beginning of the jump, peaking at approximately 75% of the jump duration and decreasing to 0 at take-off (Han et al., 2013). The force profile was approximated as a triangular, peaking at 75% of the jump force impulse length. As reported, there is no difference in the motor program of the left and right hind legs during side jumps (Santer et al., 2005), and measurements show that the reaction force of both hind legs is practically the same (Han et al., 2013). Hence we set the magnitude of the forces representing the hind legs to be equal. The maximum reaction force at the peak of the force profile was set manually so that at take-off the simulated and real locusts would propagate the same linear distance.

Figure 2 The correlations of azimuth (A–F) and elevation (G–L) of the jump with each component of the locust body position and orientation prior to jumping (a result of the aiming manoeuvres).

Lines are the best linear fit. Framed panels (C, D, H, K) denote a significant regression (Analysis of variance of linear model, F-test, p < 0.01).

Results

Invetigation of jump trajectory control through real jumps

The locust can potentially use all six DOF of its body (translation and rotation) to control the jump trajectory. To understand the effect of the initial body state (position and orientation) on the jump trajectory we examined the correlation between each coordinate of the locust’s position and orientation prior to jump initiation with the azimuth and elevation angles of the jumps (Figs. 2A–2L). The azimuth angle had a strong linear correlation with the roll and α angles (Figs. 2C and 2D). For example, a locust jumping to the left would usually roll its body and translate its COM (by changing the α angle) to the left. The elevation angle was found to have a strong linear correlation with the pitch and β angles (Figs. 2H and 2K). For example, a locust jumping strongly upwards would usually change its pitch in a head upwards manner and translate its COM upwards through changes in the β angle. The jump trajectory parameters displayed no significant correlation with the rest of the body coordinates. For further details regarding the aiming maneuvers, see examples of time-course plots of monitored jumps in the supplementary data (Fig. S2).

Dynamic model validation

To validate the dynamic model and its underlying assumptions we examined how well the simulation was able to predict the outcome (trajectory) of recorded real jumps. The azimuth and elevation angles, compared between the real and the simulated jumps, demonstrated a strong linear correlation for both criteria (Figs. 3A and 3B), thus validating that the dynamic model indeed captures the governing principles of trajectory control. Real and simulated rotational velocities were also compared, but no correlations were found (see Fig. S3 and discussion for further details). Simulations were therefore not used as a tool to further investigate jump stability.

Figure 3 Comparing real and simulated jumps.

(A) Comparison between the azimuth of real jumps and the azimuth predicted by our simulations. (B) Comparison between the elevation of real jumps and the elevation predicted by our simulations. Solid lines indicate linear regression (Analysis of variance of linear model, F-test).

Investigation of jump trajectory control through simulated jumps

To further establish a possible role for the different coordinates defining the initial body positon and orientation in the control of the jump (beyond the above reported correlations), a set of simulated jumps was performed based on each real jump. In each simulation set one coordinate was changed from its original value through its operational range (Table 1) while the remaining coordinates were kept constant at their original value. The operational range was defined by the limits of the observed distribution of each coordinate after omitting the most extreme values (10%). Hence, it presents the typical range in which the locust may vary each coordinate in order to control the jump, and is a consequence of both behavioural and physical-mechanical constraints. To test the control of azimuth, the roll and alpha angles were independently changed, and to test elevation control, the pitch and beta angles were independently changed (Figs. 4A–4D). All parameters showed an approximately linear relation with azimuth/elevation and within each simulation set graphs were consistent in slope direction and magnitude. The mean slopes for α and β were 0.98 (Std = 0.029) and 0.99 (Std = 0.029), respectively, indicating that these angles are instrumental in controlling the jumps’ azimuth and elevation angles respectively. In contrast, roll and pitch had a much more moderate effect. The roll, with a mean slope of −0.048 (Std = 0.072), could potentially change the azimuth by up to ±5°, while the pitch, with a slope of −0.0055 (Std = 0.014), had practically no effect on the elevation angle.

Table 1 Typical operational values for each coordinate during jump aiming.

Coordinate	α (deg)	β (deg)	r (mm)	ϕ (deg)	θ (deg)a	ψ (deg)	
Maximum value	28	65	8.5	17	4	20	
Minimum value	−28	20	5	−17	−17	−20	
Notes.

a The pitch angle (θ) is negative for head-upwards rotations.

Development of rotational velocity during real jumps

As noted earlier, Cofer et al. (2010) reported that the locusts set their pitch prior to jumping according to jump elevation, moving their COM in line with the hind legs’ thrust force to minimize tumbling. This strategy will diminish tumbling caused by torques resulting from the thrust force. Tumbling, however, could also be a result of the aiming manoeuvres (see detailed explanation in the ‘Introduction’) prior to the thrust force initiation: once the locust detects a threat and decides to jump away, it begins the aiming manoeuvres, during which the jump is triggered. Only then do the hind legs start to extend and exert force on the body until take-off. To investigate whether rotational instability at take-off is a result of thrust exerted by the hind legs or of the earlier aiming manoeuvres, the rotational velocities at jump initiation and at take-off were compared (Figs. 5A–5C). We found that rotational velocities at the initial jump triggering moment were already of the same scale as the rotational velocities at take-off, reaching up to 500 deg/s in yaw and roll (Figs. 5A and 5C) and 400 deg/s pitching head upwards (Fig. 5B). The yaw velocities during triggering and during take-off were uncorrelated (Fig. 5A). A strong correlation between initial and take-off roll with a slope of near 1 indicates that velocities about this axis tend to remain practically constant throughout the jump (Fig. 5C). We found that while pitch velocities at jump triggering were almost always head-upwards, pitch velocity at take-off was either head-upwards or downwards with hardly any jumps free of tumbling (Fig. 5B). The most dramatic effect of the thrust force on tumbling could be seen in jumps in which the locust tumbled head upwards prior to jumping (at triggering), but had changed its tumbling direction to head-downward by take-off. These findings indicate that although there is a significant change in rotational velocity during the jump (mainly about the pitch axis), the aiming maneuvers prior to thrust force initiation also have an important role in the development of rotational velocity.

Figure 4 The effect of initial conditions on the jump trajectory.

Each graph presents data based on 33 sets of simulated jumps. Each simulation set was based on data obtained from a specific real jump. In every graph one coordinate was manipulated through its typical operational range to quantify its effect on jump azimuth or elevation: (A) Alpha; (B) Roll; (C) Beta; and (D) Pitch. One simulation set in (B) in which changes in roll led to a change of ±4° in azimuth is marked in red.

Figure 5 A comparison of the rotational velocity about the locust’s principal axes at the beginning of hind-leg extension and at take-off.

(A) Yaw velocity. (B) Pitch velocity. (C) Roll velocity. Lines denote linear regression (Analysis of variance of linear model, F-test).

The effect of rotational velocity on the timing of flight initiation in real jumps

As noted, locusts are biased to tumble head-upwards when jumping (Cofer et al., 2010). To test the hypothesis that the purpose of the tumbling bias is to reduce risk of crashing during flight initiation, we tested correlations between the angular velocity at take-off and flight initiation timing. The angular velocity was measured just prior to take-off (2.5 ms), with take-off defined as the moment the hind-leg tarsi lost ground contact, and expressed in the locust-attached coordinate system. (In all the jumps analyzed in the current study loss of ground contact by the two legs was synchronous or within less than 2 ms.) Flight initiation timing was the difference between the time when initial hind leg extension was observed and the time that the wings started to spread. There was a linear correlation between pitch velocity and flight initiation timing such that head-downwards rotations led to earlier wing opening (Fig. 6B). No such correlation was found between flight initiation and either roll or yaw rotational velocities (Figs. 6A and 6C). Interestingly, the correlation between pitch velocity and flight initiation timing was even improved when testing pitch velocity at 5, 10 or even 15 ms prior to take-off (R2 larger than 0.5 p < 0.01; see Fig. S4).

Figure 6 The timing of flight initiation as a function of rotational velocity components expressed in a locust-attached coordinate system.

(A) Yaw velocity. (B) Pitch velocity. (C) Roll velocity. Linear regression lines are shown on all three graphs. Only the pitch velocity was significantly correlated with flight initiation timing (Analysis of variance of linear model, F-test).

Discussion

In this study we proposed a single dynamic model for the locust jump trajectory control, explaining the control of azimuth, elevation and stability. The mechanisms and strategies that have been revealed are consistent with earlier reports regarding the mechanics of elevation control (Sutton & Burrows, 2008), and the locust’s behavior during escape jumps (Santer et al., 2005). The locust can potentially use all six DOF of its body to control the jump trajectory. Hence, the full six DOF trajectory of the locust body was both monitored during real jumps and simulated using our dynamic model. This allowed us for the first time to specifically and directly determine the relative importance of each of the initial (prior to the jump) locust body coordinates in the jump trajectory control. Our investigation of the full parameter-range and parameter-combination space suggests that it is the α and β angles that are instrumental in controlling the jumps’ azimuth and elevation angles, respectively, while the rest of the body coordinates (including the body’s orientation) have little effect on the jump trajectory parameters.

As noted, real and simulated rotational velocities were compared but no correlations were found. We believe that this is due to two main reasons: (1) rotational velocities are much more sensitive to subtle changes in the initial state of the locust body than azimuth and elevation, and inaccuracies in the data initializing the simulations were too large to enable accurate prediction of rotational velocities; and (2) the locust body in our model is rigid and does not allow the abdominal flexion that may contribute to jump stabilization (Cofer et al., 2010). Simulations were therefore not used as a tool to further investigate jump stability.

Rotational instability and angular velocities are byproducts of practically all locust jumps, with important and potentially undesirable effects on flight initiation. At the beginning of the jump, at the moment the hind legs start to extend, the locust body had already accumulated rotational velocity generated during the aiming manoeuvres (Fig. 2). As we observed, aiming manoeuvres resulted in pitch rotations that were usually head-upwards. This is in accordance with the report by Cofer et al. (2010) for tumbling at take-off, but it also shows that the head-upwards rotation bias exists even before the hind legs’ thrust is initiated. This is important because different sources of jump instability might require different means for controlling it. In addition, we noted that the timing of wing opening and flight initiation was strongly dependent on pitch angular velocity. Our study consequently supports the hypothesis correlating jump stability to successful flight initiation (Cofer et al., 2010). It also indicates that the locust is sensitive to pitch angular velocity throughout the jump (and can differentiate it from the yaw and roll rotations) and that this sensory input is coupled to activation of the flight motor pattern in a yet to be explored manner (see Camhi, 1969; Pond, 1972; Reichert, 1993; Taylor, 1981 and references within, for the role of sensory inputs, including those related to pitch, during flight).

As noted earlier, Santer et al. (2005) reported that no bilateral differences in the motor programs of the left and right hind legs correlated with jump trajectory. Hind-leg asynchronous action was also reported to have no effect on jump elevation (Sutton & Burrows, 2008). The two hind legs could also have a differential effect on jump trajectory as a simple result of asynchronous loss of ground contact: prior to and during sideways jumps the locust body translates and rotates to the side, resulting in a different distance between ground contact and the TC joint of the two legs. The leg furthest from the jump direction therefore loses ground contact earlier, resulting in a short time-period during which only one leg (that still in contact with the ground) exerts forces and torques on the body. In all the jumps analysed in the current study this time difference between the loss of ground contact of the two legs was shorter than 2 ms. Based on the force profile produced by each leg throughout the jump (Han et al., 2013), the magnitude of the thrust force during the final 2 ms period is very small and diminishing. This becomes more negligible still when comparing the time-period in which only one leg produces thrust to the much longer period in which both legs exert much larger thrust. We therefore conclude that hind-leg asynchronization has a negligible effect on trajectory control and jump stability.

The locust continues to serve as an important inspiration for the development of small jumping robots (Chen et al., 2011; Kai et al., 2012; Kovač et al., 2011; Nguyen & Park, 2012). In many robotics applications and tasks there is an advantage to designing multimodal robots, capable of multiple locomotion modes. However, these introduce new challenges related to the control mechanisms and integration between modes. A locust-inspired jumping-flying robot will encounter the need to perform an efficient transition from a ballistic trajectory (jumping) to flapping-flight. Our current study suggests that the locust utilizes control and stabilization mechanisms that are based on the timing of wing-spreading (in addition to the aerodynamics of the flapping wings). Development of a bio-inspired robot based on our findings is currently underway (Zaitsev et al., 2015a; Zaitsev et al., 2015b; G Kosa & A Ayali, 2016, unpublished data) and will provide further opportunities to evaluate the contribution and importance of the presented mechanisms to flight initiation.

Supplemental Information

Figure S1 Definition of coordinate systems for kinematic analysis

The positions of markers m1, m2, m3 obtained from video analysis are expressed in the video coordinate system (xv, yv, zv). The pronotum system (xp, yp, zp) is set according to the marker positions. The ground system (xg, yg, zg) is set according to the contact points of the hind legs with the ground (Gl, Gr). The locust system (xl, yl, zl) is parallel to the pronotum system and positioned between the TC joints (TCleft, TCright). During the jump, the locust position is the difference between the origins of the locust and ground systems expressed in spherical coordinates (α, β, r).

Click here for additional data file.

Figure S2 Time-course of two jumps from the beginning of the aiming manoeuvres till takeoff

Each jump is represented by two graphs: the upper graph describes the position of the locust and the bottom graph describes the locust’s orientation. Two vertical dashed lines mark important events during the jump: The left line marks the moment when the hind legs start exerting thrust on the body. This moment was measured by noting the first frame in which the hind legs started to extend in each jump video. The right line marks the moment when the hind legs’ thrust force ends. This moment was measured by noting the frame in which the hind legs lost contact with the ground in each jump video. An image of the locust and the six coordinates describing its location and orientation are presented in figure S2C. A. First jump- The aiming manoeuvres begin about 20 milliseconds before the thrust force begins. During this phase the locust’s position changes only by changing the α angle between approximately 4 degrees at the beginning of the aiming manoeuver to approximately 20 degrees when the thrust starts. Meanwhile, during the same phase the locust changes its orientation: The pitch angle (θ) almost does not change; The yaw angle (φ) changes from approximately 9 degrees to the right to 0 degrees. The roll angle (ψ) changes from approximately 2 degrees to the left to 10 degrees to the left. Notice that the average velocity in which the roll was changed is approximately 400 degrees per second. In the second phase, the thrust phase, which takes place from the beginning of the thrust application till takeoff, the main change in the locust’s position is its propagation, which can be seen in the rise of r at approximately 30 mm till takeoff. During this phase also the orientation of the locust changes: The pitch and yaw angles (θ and ψ, respectively) continue changing in the same velocity till about 10 milliseconds before takeoff, when their velocity starts to reduce. The roll (ψ) on the other hand, changes direction and develops a higher velocity than in the previous phase. The change in roll in this phase is probably the result of torques produced by the thrust force. B. Additional jump time-course. C. Locust model with coordinate notations.

Click here for additional data file.

Figure S3 Comparison of rotational velocity at take-off between real and simulated jumps

(A) Yaw velocity. (B) Pitch velocity. (C) Roll velocity. Please note that the x and y axes are not in the same scale.

Click here for additional data file.

Figure S4 The timing of flight initiation as a function of pitch velocity measured 15 msec before take-off

The lines denote linear regression (Analysis of variance of linear model, F-test)

Click here for additional data file.

File S1 Click here for additional data file.

The authors are grateful to Gal Ribak and his lab members (Dept. of Zoology, Tel Aviv University) for the use of high speed video cameras and for lively discussions.

Appendix

Thrust direction exerted by the locust leg according to model analysis

The locust hind leg was modeled as two rigid rods representing the femur and tibia, connected by a revolute joint (Fig. A1A). The contact of the tibia with the ground and the connection between the femur and the locust body are both modeled as spherical joints with no torque applied at them. Friction and gravity are ignored and the leg segments are assumed to be massless. The force exerted by the leg is analyzed through a free body diagram (Fig. A1B) of the leg; because the segments are assumed massless, all the sum of forces and torques in each segment must be zero. Because free body diagrams are usually planar, we wish to emphasize that all the vectors in Figs. A1A and A1B are three-dimensional.

Figure A1 The locust hind leg model.

(A) Model representation of the locust hind legs. (B) Free body diagram of the hind leg.

Sum of forces on the tibia: (A1) F¯j+F¯r=0.

Sum of forces on the femur: (A2) −F¯j+−F¯b=0.

Combining Eqs. (A1) and (A2): (A3) F¯r=F¯b.

Sum of torques on the tibia around the femuro-tibial joint: (A4) −r¯ti×F¯r−M¯=0.

Sum of torques on the femur around the femuro-tibial joint: (A5) r¯fe×−F¯b+M¯=0.

Summing Eqs. (A4) and (A5) and plugging into Eq. (A3): (A6) F¯b×r¯ti+r¯fe=F¯b×r¯th=0⇒F¯b∥r¯th.

Equation (A6) concludes that F¯b and r¯th are parallel, or in other words, that the thrust force produced by a hind leg is always parallel to the line connecting the tibia’s ground contact point with the connection of the femur with the body.

F¯j Force at the femuro-tibial joint

F¯r Ground reaction force

F¯b Force exerted on body by the hind leg

M¯ Muscle produced torque in the femuro-tibial joint

r¯ti Vector from distal to proximal ends of the tibia;

r¯fe Vector from distal to proximal ends of the femur

r¯th Vector from the distal end of the tibia to the proximal end of the femur

Additional Information and Declarations

Competing Interests

Author Contributions

Data Availability

The authors declare there are no competing interests.

Omer Gvirsman conceived and designed the experiments, performed the experiments, analyzed the data, wrote the paper, prepared figures and/or tables, reviewed drafts of the paper.

Gabor Kosa and Amir Ayali conceived and designed the experiments, wrote the paper, reviewed drafts of the paper.

The following information was supplied regarding data availability:

All data has been supplied in the text or as a Supplementary File. Any further data required (e.g., video files) can be supplied upon request.

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
