# Peer review of "Dynamics and stability of directional jumps in the desert locust"

_PeerJ, doi:10.7717/peerj.2481_

## Round 0.1 · original submission · Minor Revisions

Please revise your manuscript according to the comments of the reviewers.

Reviewer 1 ·

Basic reporting

No Comments

Experimental design

1. Is there any reason why only female locusts were chosen for the experiment? Is the conclusion representative and statistically significant?

2. Is there any difference observed in mechanisms participating in the jump trajectory control caused by variant weight of the locusts?

Validity of the findings

1. Page 7, line 203: Please add the comparison of real and simulated rotational velocities in the supplementary information.

2. Page 23, Figure 3: Is there any explanation/speculation why the correlation of azimuth looks a little bit better than that of elevation?

Additional comments

The mechanisms underlying in the jump trajectory control of jumps by adult desert locusts have been systematically and specifically investigated in this study. A single dynamic model has been proposed to simulate the full six DOF trajectory of the locust body, in parallel with that monitored during real jumps, in order to explain the control of azimuth, elevation and stability. For the first time, the relative importance of each initial locust body coordinate in the jump trajectory control has been directly determined. This study is well-designed and the conclusions are reasonably supported by the experimental data analysis and discussion, therefore is recommended for publishing after some minor revisions.

Reviewer 2 ·

Basic reporting

Formulas and fiigures should be described in a more detailed level, the meaning of letter symbols for physical quantities or statistics, even though it was very common in physicists or MATLAB users. .
The use of Arabic numerals, is not an error itself, however, they should never be used at the beginning of sentences, such as “Overall 50 jumps...”.

Experimental design

Your monitoring and analysis on jump trajectory was quite complete, covering all basic factors. Good job in personal view. However, your model was still little far from jumping robot. A more accurate simulation model can make your simulation fit the real data well, before it was used in bio-inspired miniature jumping robot as a kind of important reference.

Validity of the findings

If the data did not apply to linear fitting, just show the scatter plot.
In the mechanics analysis, please sign the directions of position vectors.

Your video monitoring was precise enough in terms of both hardware and methods. Moreover, statistically, your research was data-efficient. The analysis of data was very careful, and richer analysis would make the experiments more valuable.

Additional comments

Admirably, your study was founded on adequate workload. Maybe you can discuss several potential methods to make the overall data model more specific.

·

Basic reporting

The article meets the standards.

Experimental design

The experimental question and design are appropriate.

Validity of the findings

The findings are valid and are supported by the results.

Additional comments

This is a strong paper that combines an experimental and computational approach to resolve issues about how locusts control their jumps. The paper describes experiments in which high-speed videos of locust jumps were used to determine how the animal’s initial body orientation affected the linear and angular velocities of its jump. The paper shows experimentally that two initial angles individually determine the elevation and azimuthal angles of the jump. It also shows computationally that these two angles are sufficient.
One issue that affects several parts of the paper needs to be clarified. It is the nature of the “aiming maneuvers” that the locust makes before the jump and their subsequent effect on the direction and stability of the jump. The section beginning at line 230 needs to describe the “aiming maneuvers” much more clearly and completely. Line 235 states that “the rotational velocities at the initial jump triggering moment were already of the same scale as the rotational velocity at takeoff, reaching up to 500 deg/sec in yaw and roll.” Surely these high velocities are produced by the jump itself, and are not produced by aiming maneuvers that are still occurring at the moment the jump begins. In this reviewer’s experience, “aiming maneuvers” are postural movements that adjust the animal’s joint and body angles before the jump. The jump is produced by an explosive extension of the legs that generates a force of nearly 12 N that is applied only during the initial phase of leg extension. If the force vector is not collinear with the center of mass (COM), torques will result that cause tumbling. Aiming maneuvers can increase or decrease these torques by increasing or decreasing the angle between the force vector and the direction of the COM.
Minor Comments
Line 128. A more explicit description is needed of how these observations led to the plotted angular measures.
Line 146. How was linear velocity measured or calculated?
Line 192. Reverse for causality: The azimuth angle had a strong linear correlation with the roll and alpha angles.
Line 193. The authors should show the time-course of the jump with a plot of the angles vs time after start of the jump.
Line 193. When does this change in alpha occur? Before or after jump onset or before or after take-off?
Line 194. Again, reverse: Elevation angle is correlated with pitch and beta angles.
Line 194. Should be "a locust that jumped". The jumping occurred after the animal changed its pitch and COM.
Line 197. Again, causality: the jump trajectory did or did not correlate with the body coordinates measured before the jump.
Line 235. This is unclear. Tumbling results from torques around the body axes created by the leg thrusting forces. The aiming maneuvers determine the direction of the leg thrust vector. If the force vector passes through the COM, no torques are created and tumbling will not occur. If the thrust vector does not pass through the COM, torques will develop that cause tumbling. Are you saying that during the aiming maneuver, the force vector fails to align with the COM?
Line 248. This is vague and unclear. The only way for the angular velocities to change before take-off is if the direction of the force vector changes relative to the COM. Cofer et al suggested that that could happen if contraction of the longitudinal body muscles changed the position of the COM relative to the force vector during the jump.
Line 293. If "the hind legs just started to extend (line 288), the thrust has just begun, contradicting line 292 "even before the hind legs' thrust is initiated". Moreover, the thrust forces are many times greater than those that produce the aiming movements. The pitch rotations that result from the aiming movements do so because the aiming movements caused the thrust vector to move off the COM.

---

## Round 0.2 · accepted · Accept

Your revised manuscript has been carefully reviewed. It reaches the level of publication.